# Occurrence and Molecular Characterization of *Cryptosporidium* spp. in Dairy Cattle and Dairy Buffalo in Yunnan Province, Southwest China

**DOI:** 10.3390/ani12081031

**Published:** 2022-04-15

**Authors:** Ying-Wen Meng, Fan-Fan Shu, Li-Hua Pu, Yang Zou, Jian-Fa Yang, Feng-Cai Zou, Xing-Quan Zhu, Zhao Li, Jun-Jun He

**Affiliations:** 1Key Laboratory of Veterinary Public Health of Yunnan Province, College of Veterinary Medicine, Yunnan Agricultural University, Kunming 650201, China; m1187264870@126.com (Y.-W.M.); shuff1227@163.com (F.-F.S.); plh1995815@163.com (L.-H.P.); jsc315@163.com (J.-F.Y.); zfc1207@vip.163.com (F.-C.Z.); xingquanzhu1@hotmail.com (X.-Q.Z.); 2State Key Laboratory of Veterinary Etiological Biology, Key Laboratory of Veterinary Parasitology of Gansu Province, Lanzhou Veterinary Research Institute, Chinese Academy of Agricultural Sciences, Lanzhou 730046, China; zouyangdr@163.com; 3College of Veterinary Medicine, Shanxi Agricultural University, Taigu 030801, China; 4State Key Laboratory of Conservation and Utilization of Bio-Resources in Yunnan, Center for Life Science, School of Life Sciences, Yunnan University, Kunming 650500, China

**Keywords:** *Cryptosporidium* spp., cattle, occurrence, subtype, Yunnan Province

## Abstract

**Simple Summary:**

*Cryptosporidium* spp. are important gastrointestinal pathogens of humans and animals, causing diarrheal diseases. Cattle are considered as one of the main reservoirs of *Cryptosporidium* for humans. We first report the occurrence of *Cryptosporidium* spp. in dairy cattle (14.7%, 65/442) and dairy buffalo (1.1%, 3/258) in Yunnan Province of China. The results of this study suggest that divergent *Cryptosporidium* spp. (such as *C. andersoni*, *C. bovis*, *C. ryanae*, and *C. parvum*) can be found in asymptomatic dairy cattle and dairy buffalo in Yunnan, China. The IIdA18G1 subtype of *C. parvum*, which infects humans and other animals, was also found in this study. Thus, attention should be paid towards preventing the transmission of *Cryptosporidium* spp. in cattle and humans in Yunnan Province.

**Abstract:**

*Cryptosporidium* spp. are important foodborne and waterborne pathogens in humans and animals, causing diarrheal diseases. Cattle are one of the reservoirs of *Cryptosporidium* infection in humans. However, data on the occurrence of *Cryptosporidium* spp. in cattle in Yunnan Province remains limited. A total of 700 fecal samples were collected from Holstein cows (*n* = 442) and dairy buffaloes (*n* = 258) in six counties of Yunnan Province. The occurrence and genotypes of *Cryptosporidium* spp. were analyzed using nested PCR and DNA sequencing. Furthermore, the *C. andersoni* isolates were further analyzed using multilocus sequence typing (MLST) at four gene loci (MS1, MS2, MS3, and MS16), and the *C. parvum* isolate was subtyped by 60-kDa glycoprotein (*gp*60) loci. The occurrence of *Cryptosporidium* spp. in Holstein cows and dairy buffaloes was 14.7% (65/442) and 1.1% (3/258), respectively. Of these positive samples, 56 Holstein cow samples represented *C. andersoni*, four Holstein cow samples represented *C. bovis*, three Holstein cow samples represented *C. ryanae*, and one represented *C. parvum*. Meanwhile, only three dairy buffalo samples represented *C. ryanae*. MLST analysis of subtypes of *C. andersoni* detected four subtypes, including A5A4A2A1 (*n* = 7), A4A4A4A1 (*n* = 7), A1A4A4A1 (*n* = 2), and A4A4A2A1 (*n* = 1). One *C. parvum* isolate was identified as the IIdA18G1 subtype. These results revealed the high occurrence and high genetic diversity of *Cryptosporidium* spp. in Holstein cows in Yunnan Province, enriching the knowledge of the population genetic structure of *Cryptosporidium* spp. in Yunnan Province.

## 1. Introduction

*Cryptosporidium* spp. are important apicomplexan parasites, causing moderate-to-severe diarrhea in many vertebrates [1]. This parasite can be transmitted through the fecal–oral route and direct contact [2]. Traditionally, cattle are considered as the major reservoir for human and animal infection by *Cryptosporidium* spp. [3,4]. Many cryptosporidiosis cases have been reported in humans and calves around the world, even causing adverse effects on the dairy industry and public health [5,6,7].

At present, approximately 40 *Cryptosporidium* species and 100 genotypes have been identified, of which *C. andersoni*, *C. bovis*, *C. ryanae*, and *C. parvum* are the four most prevalent species that are responsible for cattle infection, and *C. parvum* is the main zoonotic species [8]. Numerous studies have shown that these four *Cryptosporidium* spp. present an age-related distribution in dairy cattle, with *C. parvum* being the predominant species in pre-weaned calves, causing significant diarrhea, *C. bovis* and *C. ryanae* usually infect post-weaned calves and yearlings, without obvious diarrhea, whereas *C. andersoni* is usually found in adult cattle with poor production performance [9,10,11]. Currently, multilocus sequence typing (MLST) is a high-efficiency tool for typing *C. muris* and *C. andersoni*, while 14 MLST of *C. andersoni* subtypes have been reported in cattle [12,13,14,15]. At present, 10 MLST subtypes of *C. andersoni* have been recognized in cattle in China, of which A1A4A4A1 is a preponderant subtype [13,14,15,16]. Furthermore, the 60-kDa glycoprotein (*gp*60) is one of the prevalent typing gene markers for *C. parvum* genotypes [17]. Thus far, 15 subtype families of *C. parvum* have been reported, including IIa to IIi and IIk to IIp, in which the IIa and IId families are defined as zoonotic pathogens [17,18].

*Cryptosporidium* spp. infections in cattle have been reported in more than 24 provinces, autonomous regions, and municipalities of China [3,8,19]. In a small-scale survey of Yunling cattle in Yunnan Province, 0.77% cattle were infected with *C. andersoni* and *C. ryanae* [20]. However, little is known of the occurrence and subtypes of *Cryptosporidium* spp. infections in dairy cattle and dairy buffalo in Yunnan Province. Thus, the objectives of the present study were to perform a molecular epidemiological survey of the occurrence and subtypes of *Cryptosporidium* spp. in Holstein cows and dairy buffaloes to assess the zoonotic potential of *Cryptosporidium* spp. in cattle in Yunnan Province.

## 2. Materials and Methods

### 2.1. Sample Collection

In total, 700 fecal samples were collected from seven Holstein cow farms (*n* = 442) in four counties (Yiliang, Eryuan, Weishan, and Binchuan) and five crossbred dairy buffalo farms (*n* = 258) in two counties (Heqing, Tengchong) in Yunnan Province from June 2019 to August 2020 (Figure 1). All cattle of each farm were sampled. Each sample was collected from the rectum individually, using disposable plastic gloves, and was transferred separately into a disposable self-sealing bag with accurate label information. The animals were divided into four age groups, including pre-weaned calves (less than 3 months), post-weaned calves (3–12 months), heifers (13–24 months), and adult cattle (more than 24 months). All cattle were clinically normal and without obvious signs of diarrhea. All samples were stored in 2.5% potassium dichromate and sent to the laboratory for DNA extraction.

### 2.2. DNA Extraction

Each fecal sample was washed with sterile water and centrifuged at 2000× *g* for 15 min to remove potassium dichromate. The extraction of genomic DNA from individual samples was performed using a commercial E.Z.N.A Stool DNA Kit (Omega Bio-tek Inc., Norcross, GA, USA, http://www.omegabiotek.com/ (accessed on 5 July 2020)), following the manufacturer’s recommended procedures. The genomic DNA was stored at –20 °C until its use in PCR amplification.

### 2.3. PCR Amplification

*Cryptosporidium* spp. species are determined by a nested PCR targeting the small subunit (*SSU*) rRNA gene [17]. The *C. andersoni*-positive samples were subsequently analyzed in four minisatellite/microsatellite targets, including MS1, MS2, MS3, and MS16 loci, according to previous studies [12,13]. The *C. parvum*-positive samples were further sub-typed according to the 60-kDa glycoprotein (*gp*60) gene [21]. Each specimen was set with two technical replicates at each genetic locus, and visualized by agarose gel electrophoresis.

### 2.4. Sequence Analysis and Phylogenetic Tree

All secondary PCR products were sent to Sangon Biotech (Sangon, Shanghai city, China) for bidirectional sequencing using an ABI 3170 Genetic Analyzer (Applied Biosystems, Foster City, CA, USA) to obtain the sequences of the PCR products. Software ChromasPro 2.1.5.0 (http://technelysium.com.au/ChromasPro.html (accessed on 10 September 2020)), BioEdit 7.1 (http://www.mbio.ncsu.edu/BioEdit/bioedit.html (accessed on 10 September 2020)), and ClustalX 2.1 (http://clustal.org (accessed on 21 September 2020)) were used to assemble and align the nucleotide sequences. The *Cryptosporidium* species were determined by comparison with relevant reference sequences in GenBank. The *C. andersoni* and *C. parvum* subtypes were named according to the established nomenclature [12,22]. The phylogenetic tree was constructed using the maximum-likelihood method to assess the phylogenetic relationship among *Cryptosporidium* spp. genotypes and subtypes using MEGA 7.0 (http://www.megasoftware.net/ (accessed on 15 October 2020)) based on substitution rates calculated by the general time-reversible model. The reliability of the data was assessed using bootstrapping with 1000 replicates. 

### 2.5. Statistical Analysis

The differences in occurrence of *Cryptosporidium* spp. between regions and age groups were calculated using the Fisher’s exact test with the software Statistical Product and Service Solutions 20.0 (SPSS 20.0) (IMB Corporation, Armonk, NY, USA). Differences were considered significant at *p* ≤ 0.05. Odds ratios (OR) with 95% confidence intervals (CI) were generated using the software SPSS 20.0 to assess the strength of location and age factors in cattle.

## 3. Results

### 3.1. Occurrence of Cryptosporidium spp

Of the 442 fecal samples collected from Holstein cows on seven farms, 14.7% (*n* = 65) were positive for *Cryptosporidium* spp., with infection rates ranging from 0% (0/18) to 38.0% (38/100) (Table 1). Farm Yiliang-2 in Kunming city had a significantly higher occurrence rate than the other farms (*p* < 0.01). Among the samples from the four age groups, the occurrence rates of *Cryptosporidium* spp. varied from 12.9% to 50%. The difference in *Cryptosporidium* occurrence among the four age groups was not significant (*p* = 0.088). Of the 258 fecal samples analyzed, only 1.1% (*n* = 3) of samples from diary buffaloes at Tengcong-1 farm were positive for *Cryptosporidium* (Table 2). The difference in *Cryptosporidium* occurrence among the five farms was not significant. Among samples from the four age groups, only pre-weaned calves were detected with *Cryptosporidium*, and the difference in *Cryptosporidium* occurrence was not significant between age groups (*p* = 0.281).

### 3.2. Genotyping of Cryptosporidium spp

In this study, four *Cryptosporidium* species, namely *C. andersoni* (*n* = 56), *C. bovis* (*n* = 5), *C. ryanae* (*n* = 2), and *C. parvum* (*n* = 1), in Holstein cows, and one species, *C. ryanae* (*n* = 3), in dairy buffaloes, were identified. Regarding the Holstein cow farms, except for farm Binchuan-2, *C. andersoni* was observed on all farms, while *C. bovis*, *C. ryanae*, and *C. parvum* were also detected on more than three farms (Table 1). In terms of the dairy buffaloes, *C. ryanae* was the only species observed on the Tengchong-1 farm (Table 2).

In Holstein cows, two or more *Cryptosporidium* species were found in age groups other than pre-weaned calves, and only one species, *C. bovis*, was detected in pre-weaned calves. In addition, *C. ryanae* was found in heifers and adult cattle, and *C. parvum* existed in post-weaned calves. Similarly, *C. ryanae* was identified in pre- and post-weaned dairy buffalo calves. The *Cryptosporidium* spp. sequences found in Holstein cows were identical to reference sequences JN400881 (*C. andersoni*), MT950118 (*C. bovis*), JN400880 (*C. ryanae*), and MF671870 (*C. parvum*). In contrast, among the sequences obtained from dairy buffaloes, one sequence was identical to reference JN400880 (*C. ryanae*), and another two sequences were highly similar to reference JN400880 (*C. ryanae*), with one nucleotide substitution. The phylogenetic trees showed that the sequences of *Cryptosporidium* genotypes were clustered with their reference sequences (Figure 2).

### 3.3. Subtyping of Cryptosporidium andersoni and Cryptosporidium parvum

All the *C. andersoni*-positive samples were further subtyped by MLST using four loci (MS1, MS2, MS3, and MS16). However, only 17 of 56 *C. andersoni* isolates were successfully subtyped by the four loci, forming four MLST subtypes. Of these MLST subtypes, A5A4A2A1 (*n* = 7) and A4A4A4A1 (*n* = 7) were most prevalent on dairy cattle farms in Yiliang-2, followed by subtypes A1A4A4A1 (*n* = 2) and A4A4A2A1 (*n* = 1) observed on Eryuan farm. Furthermore, only one subtype (IIdA18G1) was successfully sequenced, based on a *C. parvum*-positive sample, at the *gp*60 gene locus (Table 1).

## 4. Discussion

This is the first report of the detection of *Cryptosporidium* spp. in Holstein cows and dairy buffaloes in Yunnan Province of China, with occurrences of 14.7% and 1.1%, respectively. The occurrence rates of these pathogens were higher than those found in Yunling cattle in a previous study [20]. The occurrence of *Cryptosporidium* spp. in Holstein cows was similar to most previous studies conducted in China, with occurrence rates of 8.5% to 21.2% [9,10,15,23,24,25,26,27,28]. In contrast, the occurrence of *Cryptosporidium* spp. in dairy buffaloes was lower than that found in Hunan Province (23.8%) [29] and the average rate for China (15.5%) [3]. The reasons for the different occurrence rates remain unclear, but the geography, age distribution of samples, the timing of sample collection, and sample sizes could be the contributing factors.

The four common *Cryptosporidium* species were identified in the present study. In the Holstein cows, *C. andersoni* was the predominant species, which is consistent with previous findings in China, India, and Brazil [9,14,24,30,31,32]. Although the number of samples from the pre-weaned and post-weaned calves, and heifers age groups was small, the age-related trend of *Cryptosporidium* spp. is similar to other studies; for example, in pre/post-weaned calves, *C. bovis* and *C. parvum* have been detected, while the occurrence of *C. andersoni* and *C. ryanae* was reported to gradually increase with increasing age [25,29,33]. Thus, further studies are needed to reveal the age-related pattern of *Cryptosporidium* spp. in Yunnan, China. Only one species, *C. ryanae,* was detected in the calves of dairy buffaloes, which is similar to previous data for buffaloes in Nepal and Egypt [34,35].

A high diversity of MLST subtypes of *C. andersoni* was detected in this study. Thus far, six cattle-associated *C. andersoni* subtypes have been reported in China [8], while four subtypes were detected in Yunnan Province. Among these subtypes, one MLST subtype (A4A4A4A1) was the prevalent subtype in Holstein cows, which is consistent with previous findings that these isolates are most common in cattle. In addition, three subtypes (A1A4A4A1, A5A4A4A1, and A4A4A2A1) detected in this study have also been reported in dairy cattle, beef cattle, and Qinchuan cattle, respectively [13,15].

*Cryptosporidium parvum* IIdA18G1 was first identified in Holstein cows in Yunnan Province. *Cryptosporidium parvum* is a zoonotic species occurring in pre-weaned calves. In most areas, *C. parvum* IIa subtypes are the primary factors causing diarrhea in calves [36,37,38]. In contrast, IId subtypes are commonly found in calves in China [2,39]. Until now, five subtypes of *C. parvum*, IIdA14G1, IIdA15G1, IIdA17G1, IIdA19G1, and IIdA20G1, have been identified in dairy cattle in China [3,5,19]; three of these five *C. parvum* subtypes (IIdA15G1, IIdA9G1, and IIdA20G1) have been reported as highly pathogenic subtypes responsible for a cryptosporidiosis outbreak in China [5,6,40]. Moreover, IIdA15G1 and IIdA19G1 are the dominant subtypes in dairy cattle in China [19], whereas the IIdA18G1 subtype has been reported in yaks in China [41], in dairy cattle in Serbia [42], in lambs in Spain [43,44], and in humans in the United kingdom and Kuwait [45,46]. Therefore, the presence of this subtype in Holstein cows imposes a potential threat of cryptosporidiosis transmission.

## 5. Conclusions

The present study revealed the high occurrence of *Cryptosporidium* spp. infection in Holstein cattle from Yunnan Province, southwest China. The IIdA18G1 subtype of *C. parvum* represents a significant public health concern for humans, as a cattle attendant may come into contact with infected cattle without any biological protection. The four MLST subtypes (A5A4A4A1, A4A4A4A1, A4A4A2A1, and A1A4A4A1) of *C. andersoni* were mainly reported in cattle, which might have long-term adverse effects on the dairy cattle industry. In further studies, a higher number of samples from young animals will illuminate the complete picture of the occurrence of *Cryptosporidium* species in relation to particular age groups. This will allow a better understanding of *Cryptosporidium* infections in dairy cattle from Yunnan, and it will be interesting to know which age groups are important reservoirs of zoonotic and/or pathogenic species.

## Figures and Tables

**Figure 1 animals-12-01031-f001:**
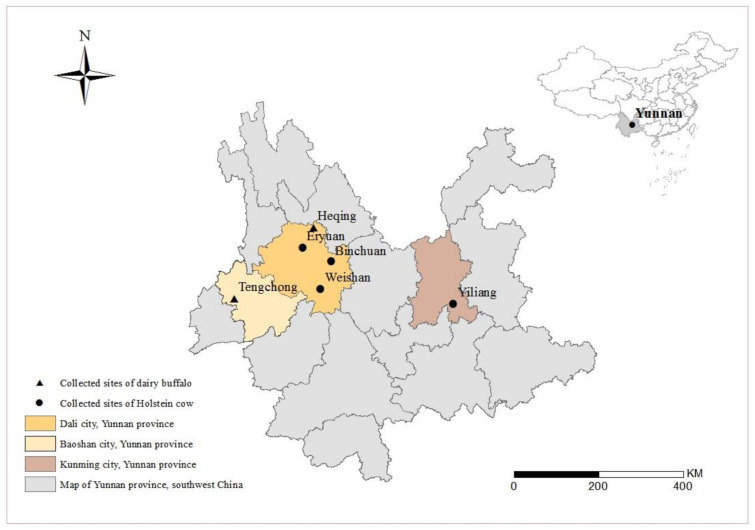
Map of sampling sites for Holstein cows and dairy buffaloes in Yunnan Province, China.

**Figure 2 animals-12-01031-f002:**
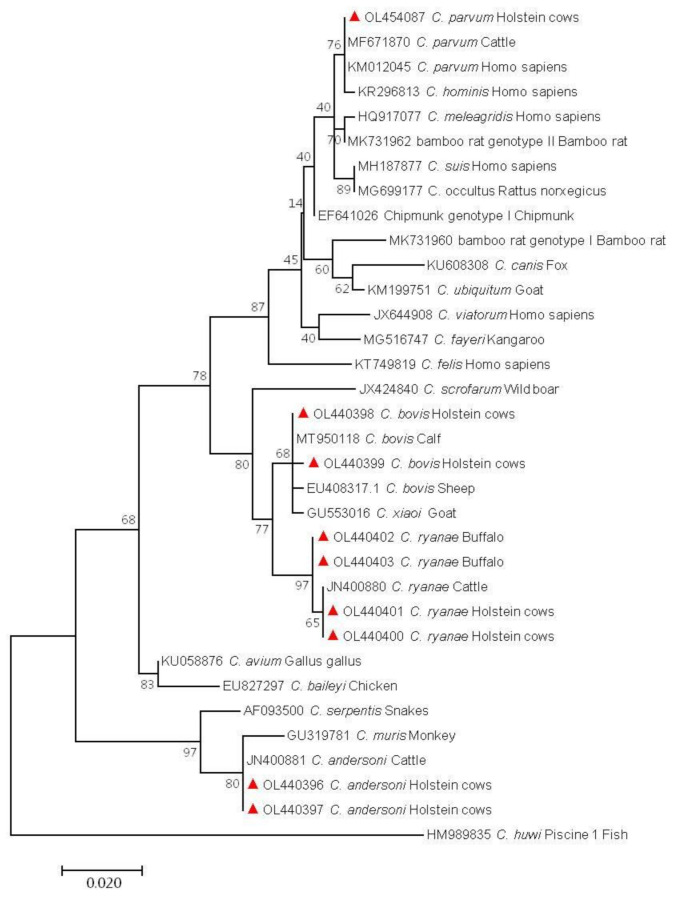
Phylogenetic tree depicting evolutionary relationships among *Cryptosporidium* spp. sequences at the *SSU* rRNA locus. ▲: Sequence obtained in this study.

**Table 1 animals-12-01031-t001:** Occurrence and identification of *Cryptosporidium* spp. in Holstein cows in Yunnan Province, China.

Location and Age	No. Positive/Sample Size	Occurrence (%)	*p*-Value	OR(95% CI)	*Cryptosporidium* Species	*C. andersoni* Subtype	*C. parvum* Subtype
Sampling sites							
Yiliang-1 Kunming city	15/148	10.1	0.076	5.64 (0.73–43.8)	*C. andersoni* (9), *C. bovis* (4), *C. ryanae* (1), *C. parvum* (1)	-	IIdA18G1(1)
Yiliang-2 Kunming city	38/100	38.0	0.000	30.65 (4.06–231.08)	*C. andersoni* (36), *C. ryanae* (1), *C. bovis* (1)	A5A4A2A1 (7) A4A4A4A1 (7)	-
Eryuan Dali city	6/45	13.3	0.048	7.69 (0.89–66.57)	*C. andersoni* (6)	A4A4A2A1 (1) A1A4A4A1 (2)	-
Weishan Dali city	1/51	1.9	-	Reference	*C. andersoni* (1)	-	-
Binchuan-1 Dali city	3/58	5.1	0.621	2.73 (0.28–27.08)	*C. andersoni* (3)	-	-
Binchuan-2 Dali city	0/18	0	-	-	-	-	-
Binchuan-3 Dali city	2/22	9.0	0.214	5.00 (0.43–58.28)	*C. andersoni* (1), *C. ryanae* (1)	-	-
Age (months)							
Pre-weaned calves (<3)	2/4	50.0	0.088	6.72 (0.92–48.97)	*C. bovis* (2)	-	-
Post-weaned calves (3–12)	11/67	16.4	0.449	1.32(0.64–2.72)	*C. andersoni* (10), *C. parvum* (1)	A5A4A2A1(3)	IIdA18G1(1)
Heifers (13–24)	9/39	23.0	0.085	2.02(0.90–4.54)	*C. andersoni* (6), *C. ryanae* (1), *C. bovis* (2)	A4A4A4A1(1) A4A4A2A1(1) A1A4A4A1(2)	-
Adults (>24)	43/332	12.9	-	Reference	*C. andersoni* (40), *C. ryanae* (2), *C. bovis* (1)	A5A4A2A1(4) A4A4A4A1 (6)	-
Total	65/442	14.7			*C. andersoni* (56), *C. bovis* (5), *C. ryanae* (3), *C. parvum* (1)	A5A4A2A1 (7) A4A4A4A1 (7) A4A4A2A1 (1) A1A4A4A1 (2)	IIdA18G1(1)

No, number; CI, confidence interval; OR, odds ratio.

**Table 2 animals-12-01031-t002:** Occurrence and identification of *Cryptosporidium* spp. in dairy buffaloes in Yunnan Province, China.

Location and Age	No. Positive/Sample Size	Occurrence (%)	*p*-Value	OR(95% CI)	*Cryptosporidium* Species
Collection site					
Heqing Dali city	0/34	0	-	-	-
Tengchong-1 Baoshan city	3/133	2.2	-	-	*C. ryanae* (3)
Tengchong-2 Baoshan city	0/31	0	-	-	-
Tengchong-3 Baoshan city	0/28	0	-	-	-
Tengchong-4 Baoshan city	0/32	0	-	-	-
Age groups (months)			-	-	-
Pre-weaned calves (<3)	2/11	18.1	0.281	4.22(0.34–52.91)	*C. ryanae* (2)
Post-weaned calves (3–12)	1/20	5.0	-	Reference	*C. ryanae* (1)
Heifers (13–24)	0/21	0	-	-	-
Adults (>24) Total	0/206 3/258	0 1.1	- -	- -	- -

No, number; CI, confidence interval; OR, odds ratio.

## Data Availability

The sequences obtained in the study were deposited in GenBank under the accession numbers OL420756, OL440396–OL440403, and OL454087.

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
