# Peer review of "Occurrence and Molecular Characterization of *Cryptosporidium* spp. in Dairy Cattle and Dairy Buffalo in Yunnan Province, Southwest China"

_animals, 2022, doi:10.3390/ani12081031_

Round 1

Reviewer 1 Report

This is an interesting and simple study which describes the types of Cryptosporidium which circulates around Yunnan province in China. Using a previously published MLST protocol, the authors produce an interesting study comparing Yunnan province with other areas of the world and China. It is well written in general, and I only have minor comments which are detailed below

Line 61- this may be better as cattle rather than bovine

Line 68- you can delete species here as you have spp after Cryptosporidium

Line 74- you jump in a little quick to the MLST stuff here. Perhaps a line where you mention the previous studies and the loci used may be beneficial?

Line 106- nested PCR

Several places- particularly from lines 114-132 need Cryptosporidium and specific species putting into italics

Line 137- significant, rather than significantly

Section 3.1.- it may be useful to include the n numbers in here as well

Table 2- give you have so many negative results- so you need to include table 2? I will leave this up to the authors

Line 164- in the other three …(reword)

Line 177- subtypes by MLST using four loci …. (and then delete by MLST on line 178) – reword

Line 193- with a prevalence of …. (reword)

Line 202-203- It may work better to link these sentences if possible

Line 207-214- there are a lot of times that subtypes is mentioned- is it possible to reword for ease of reading?

Line 221- resulting in a cryptosporidiosis …. (reword)

Author Response

Response to Reviews for animals-1606913

Many thanks for the constructive comments and suggestions. We have taken all suggestions by the reviewers into consideration, and in each case, made the suggested modifications. We used the “tracked changes” mode in the WORD to show the revised/changed text in the revised manuscript. Two manuscript files are uploaded: the “clean version” as “manuscript”, and the one showing “tracked changes”. In the following, we detail our point-by-point responses to the reviewer’s comments and suggestions.Below are the specifics of the revision (reviewers’ suggestion first followed by the response). We made all our responses in blue color for clarity. The following line numbers correspond to those in the marked copy of the revised manuscript.

Responses to reviewer #1

This is an interesting and simple study which describes the types of Cryptosporidium which circulates around Yunnan province in China. Using a previously published MLST protocol, the authors produce an interesting study comparing Yunnan province with other areas of the world and China. It is well written in general, and I only have minor comments which are detailed below

RESPONSE: Many thanks for the positive comments.

  1. Line 61- this may be better as cattle rather than bovine

RESPONSE: We have added the information in Line 55, 61.

  1. Line 68- you can delete species here as you have spp after Cryptosporidium

RESPONSE: We have added the information in Line 63.

  1. Line 74-You jump in a little quick to the MLST stuff here. Perhaps a line where you mention the previous studies and the loci used may be beneficial?

RESPONSE: We have added the information and reference in Line 68.

  1. Line 106- nested PCR, Several places- particularly from lines 114-132 need Cryptosporidium and specific species putting into italics

RESPONSE: We have made corresponding revisions in lines 104 and 112-128.

  1. Line 137- significant, rather than significantly

RESPONSE: We have rewritten the section 3.1.

  1. Section 3.1.- it may be useful to include the n numbers in here as well

RESPONSE: We rewritten the section 3.1 and added n in the paragraph in lines 135-146.

  1. Table 2- give you have so many negative results- so you need to include table 2? I will leave this up to the authors

RESPONSE: Many thanks for your suggestion. As dairy buffaloes are the important cattle breeds in Yunnan Province, the investigations of Cryptosporidium spp. might be useful data. Thus, we hope to include these results in the investigation.

  1. Line 164- in the other three …(reword)

RESPONSE: We have made corresponding revisions in line 163.

  1. Line 177- subtypes by MLST using four loci …. (and then delete by MLST on line 178) – reword

RESPONSE: We have made corresponding revisions in lines 177-178.

  1. Line 193- with a prevalence of …. (reword)

RESPONSE: We have added the information in line 189.

  1. Line 202-203- It may work better to link these sentences if possible

RESPONSE: We have made corresponding revisions in lines 197-200.

  1. Line 207-214- there are a lot of times that subtypes is mentioned- is it possible to reword for ease of reading?

RESPONSE: We have made corresponding revisions in lines 203-209.

  1. Line 221- resulting in a cryptosporidiosis …. (reword)

RESPONSE: We have made corresponding revision in line 217.

Reviewer 2 Report

This study provides data on the prevalence and molecular characterization of Cryptosporidium  spp. in cattle in China.

The manuscript is overall well planned and written.

There are some points that need the authors’ attendance before the manuscript is suitable for publication. These are listed below.

Lines 70, 216: expand genus name at the beginning of sentence

Lines 113-129: Latin names in italics

Line 152: „identification“ instead of „identify“; „Cryptosporidium: in italics

Line 157: remove „species“

Line 229: „andersoni“ not „andersonic“

Author Response

Response to Reviews for animals-1606913

Many thanks for the constructive comments and suggestions. We have taken all suggestions by the reviewers into consideration, and in each case, made the suggested modifications. We used the “tracked changes” mode in the WORD to show the revised/changed text in the revised manuscript. Two manuscript files are uploaded: the “clean version” as “manuscript”, and the one showing “tracked changes”. In the following, we detail our point-by-point responses to the reviewer’s comments and suggestions.Below are the specifics of the revision (reviewers’ suggestion first followed by the response). We made all our responses in blue color for clarity. The following line numbers correspond to those in the marked copy of the revised manuscript.

Responses to reviewer #2

This study provides data on the prevalence and molecular characterization of Cryptosporidium spp. in cattle in China. The manuscript is overall well planned and written. There are some points that need the authors’ attendance before the manuscript is suitable for publication. These are listed below.

RESPONSE: Many thanks for the positive comments.

  1. Lines 70, 216: expand genus name at the beginning of sentence

RESPONSE: We have added the information in Lines 64, 211.

  1. Lines 113-129: Latin names in italics

RESPONSE: We have made corresponding revisions in lines 111-128.

  1. Line 152: „identification“ instead of „identify“; „Cryptosporidium: in italics

RESPONSE: We have made corresponding revisions in line 151.

  1. Line 157: remove „species“

RESPONSE: We have made corresponding revision in line 156.

  1. Line 229: „andersoni“ not „andersonic“

RESPONSE: We have made corresponding revision in line 225.

Reviewer 3 Report

This reseacrh “Prevalence and Molecular Characterization of Cryptosporidium spp. in Dairy Cattle in Yunnan Province, Southwest China” aims to examine the molecular prevalence and genetically characterize Cryptosporidium spp. in dairy cattle in Yunnan province, Southwest China using nested PCR and DNA sequencing. The authors report that the results revealed the high prevalence and high genetic diversity of Cryptosporidium spp. in Holstein cows in Yunnan province, which enrich the better understanding of the population genetic structure of Cryptosporidium spp., and have implications for the better control of Cryptosporidium spp in human and animals.

It is a comprehensive, easy to read and informative work with useful results to increase knowledge about the infection and epidemiology of Cryptosporidium.

Only a few minor issues that should be modified in the manuscript:

Review all scientific names and write in italics, especially section 2.4 Sequence analysis and phylogenetic tree

Line 135: “The overall prevalence of Cryptosporidium spp. was 14.7% (68/700) in Holstein cows…”, please correct these data as they are not correct.

Figure 2: please modify the figure as the sequence GU319781 C. muris Monkey is out of position.

Author Response

Response to Reviews for animals-1606913

Many thanks for the constructive comments and suggestions. We have taken all suggestions by the reviewers into consideration, and in each case, made the suggested modifications. We used the “tracked changes” mode in the WORD to show the revised/changed text in the revised manuscript. Two manuscript files are uploaded: the “clean version” as “manuscript”, and the one showing “tracked changes”. In the following, we detail our point-by-point responses to the reviewer’s comments and suggestions. Below are the specifics of the revision (reviewers’ suggestion first followed by the response). We made all our responses in blue color for clarity. The following line numbers correspond to those in the marked copy of the revised manuscript.

Responses to reviewer #3

This reseacrh “Prevalence and Molecular Characterization of Cryptosporidium spp. in Dairy Cattle in Yunnan Province, Southwest China” aims to examine the molecular prevalence and genetically characterize Cryptosporidium spp. in dairy cattle in Yunnan province, Southwest China using nested PCR and DNA sequencing. The authors report that the results revealed the high prevalence and high genetic diversity of Cryptosporidium spp. in Holstein cows in Yunnan province, which enrich the better understanding of the population genetic structure of Cryptosporidium spp., and have implications for the better control of Cryptosporidium spp in human and animals.

It is a comprehensive, easy to read and informative work with useful results to increase knowledge about the infection and epidemiology of Cryptosporidium.

RESPONSE: Many thanks for the positive comments.

Only a few minor issues that should be modified in the manuscript:

  1. Review all scientific names and write in italics, especially section 2.4 Sequence analysis and phylogenetic tree

RESPONSE: We have made corresponding revisions about the scientific names.   

  1. Line 135: “The overall prevalence of Cryptosporidiumspp. was 14.7% (68/700) in Holstein cows…”, please correct these data as they are not correct.

RESPONSE: We have corrected corresponding data in Line 135.

  1. Figure 2: please modify the figure as the sequence GU319781 C. murisMonkey is out of position.

RESPONSE: We have made corresponding revision in the figure 2.

Reviewer 4 Report

Dear authors

The manuscript “Prevalence and Molecular Characterization of Cryptosporidium spp. in Dairy Cattle in Yunnan Province, Southwest China” provides information on the presence of Cryptosporidium infections in dairy cattle and water buffaloes in a province of China where previous data is limited. Although it is well written, the techniques used are correct and the molecular results are scientifically sound, there are some important shortcomings related to the sampling:

-How was the total number of samples calculated?

-The number of samples from most age groups is very small. This can significantly influence the prevalence values observed, since it is well known that the prevalence of Cryptosporidium in most animal species diminishes with age. In addition, the term “prevalence” should not be used in these age groups but “occurrence” (i.e. the percentage of positive preweaned calves is 50%... but only 4 animals were analysed… Can you extrapolate this percentage value to all preweaned calves?).

-Only seven farms were studied. How was the sampling procedure in each farm? How many samples from each age group were collected in each farm? A careful assessment of your data indicates that no preweaned calves were samples in some farms… It is difficult to compare results between farms if the sampling protocol was different. In addition, the sampling procedure was also different in buffalo farms (the proportion of preweaned calves -0.9 (4/442)- was lower than the proportion of preweaned buffaloes -4.3-, and the proportion of postweaned calves was two times higher -15.2- than that of postweaned buffaloes -7.8). For that reason, you cannot compare prevalences/occurrences.

-It was not stated if diarrhoeic animals were sampled. It is also well known that Cryptosporidium infections (especially C. parvum infections) are more frequent in diarrhoeic animals.

-For all these reasons, you cannot complete the first objective (perform the molecular epidemiological survey of prevalence of Cryptosporidium spp. in Holstein cows and dairy buffaloes in Yunnan province).

In addition, although this manuscript provides some new information from a region where data is scarce, results are similar to other previously described in China (and not included in the discussion section). For all these reasons, this manuscript should be significantly improved for being considered for publication in “animals”.

OTHER COMMENTS

-Title: in addition to “dairy cattle” add “dairy water buffaloes”

-Simple summary should be more accessible for all readers. I think it is still too much technical for all readers.

-Lines 82-82. It is true: little is known about Cryptosporidium infections in cattle from Yunnan. But there is other study (some authors of the present manuscript are also authors of the other investigation) that is not included: Liang et al. (2021). First report of the prevalence and genetic characterization of Giardia duodenalis and Cryptosporidium spp. in Yunling cattle in Yunnan Province, southwestern China, Microbial Pathogenesis,158,105025. https://doi.org/10.1016/j.micpath.2021.105025. Please include and discuss it.

-Line 116: delete one of the dots.

-Sections 2.4 and 2.5: Scientific names should be written in italics.

-Lines 131-132: Indicate how OR and CI95% were calculated.

-Results section: This section should be revised carefully. Firstly, most results (percentages) in the text are repeated in Tables 1 and 2. Do not repeat data on Tables on the text. I recommend keeping percentages on both Tables and indicating patterns or trends in the text. Secondly, be careful with the chi-square results. In some parts it is stated that the differences are significant but they are not. For example in lines 141-144 it is stated that “prevalence in the pre-weaned calves (50.0%) was significantly higher than in post-weaned 142 calves (16.4%), heifers (23.0%), and adult cattle” but this is not completely true. The percentage of infection in preweaned calves was only significantly higher than in adults. The rest were not calculated or not included in the Tables/text. I calculated some and they were not significant. Results in lines 144-146 do not agree with those on the Table 2…  Most differences between age groups are not significant because the number of samples analysed was very small.

In addition, some p values are significant but CI95% of OR values included the number 1… So it should not be significant… Something is wrong. If you are performing a chi-square including all age groups (or considering farms) it will be significant. But if you compare age groups 2 by 2 (i.e. considering the adult group as a reference) most comparisons will not be significant. Be careful since chi-square is not suitable if you have a 2x2 contingency table and some cells have numbers under 5… You should use an exact Fisher test.

For example, if you compare calves (2/4 positives) and adults (43/332), you can see in Table 1 that p= 0.031 and OR was 6.72 (0.92-48.97). Using R:

Your results are obtained like this:

> tabla <- matrix(c(2,2,43,289), nrow = 2)

> chisq.test(tabla, correct = FALSE)

Pearson's Chi-squared test

data:  tabla

X-squared = 4.677, df = 1, p-value = 0.03057

Warning message:

In chisq.test(tabla, correct = FALSE) :

  Chi-squared approximation may be incorrect

> library(vcd)

Loading required package: grid

> oddsratio(tabla, log = FALSE)

 odds ratios for and

[1] 6.72093

> confint(oddsratio(tabla, log = F))

      2.5 %   97.5 %

/ 0.9224279 48.96958

>

Nevertheless, all this is wrong because you have two cells with less than 5. You must consider that when the number of observations expected for any of the levels is equal to or less than 5, the approximation by the chi square test is not good. Thus, you should use a fisher test (exact test)

> fisher.test(tabla)

Fisher's Exact Test for Count Data

data:  tabla

p-value = 0.08828

alternative hypothesis: true odds ratio is not equal to 1

95 percent confidence interval:

  0.4711809 94.1551091

sample estimates:

odds ratio

  6.656798 

In this case, p is > 0.05, so it is no significant.

-Line 135: it is not 68/700 (overall prevalence) but 65/442. Anyway, these data is on the Table.

-Lines 137-139: since sampling protocol was very different between farms, comparing overall infection rates has no sense. You can report that the “herd-level occurrence/prevalence” ranged from xx% to xx%.

 -Figures 1 and 2. In my opinion, phylogenetic trees are not needed (what did you use as root of both trees?). The major Cryptosporidium species found are well defined at the 18S gene.

-The discussion section is poor.

-A predominance of C. andersoni in all age groups in cattle was detected. This is very common in Chinese and in some Brazilian studies on cattle, but very different to the pattern described elsewhere. Please discuss that.

-Lines 202-204. Something is wrong here.

-IId subtypes are not the most common in cattle worldwide but IIa. Why do you think this C. parvum allelic family is predominant in China? Is this allelic family more frequent in cattle from other countries too?

Author Response

Response to Reviews for animals-1606913

Many thanks for the constructive comments and suggestions. We have taken all suggestions by the reviewers into consideration, and in each case, made the suggested modifications. We used the “tracked changes” mode in the WORD to show the revised/changed text in the revised manuscript. Two manuscript files are uploaded: the “clean version” as “manuscript”, and the one showing “tracked changes”. In the following, we detail our point-by-point responses to the reviewer’s comments and suggestions.Below are the specifics of the revision (reviewers’ suggestion first followed by the response). We made all our responses in blue color for clarity. The following line numbers correspond to those in the marked copy of the revised manuscript.

Responses to reviewer #4

Reviewer #4:

The manuscript “Prevalence and Molecular Characterization of Cryptosporidium spp. in Dairy Cattle in Yunnan Province, Southwest China” provides information on the presence of Cryptosporidium infections in dairy cattle and water buffaloes in a province of China where previous data is limited. Although it is well written, the techniques used are correct and the molecular results are scientifically sound, there are some important shortcomings related to the sampling:

RESPONSE: We thanks reviewer #4 very much for the constructive comments and suggestions.

  1. How was the total number of samples calculated?

RESPONSE: We have corrected corresponding data in section 3.1.

  1. The number of samples from most age groups is very small. This can significantly influence the prevalence values observed, since it is well known that the prevalence of Cryptosporidium in most animal species diminishes with age. In addition, the term “prevalence” should not be used in these age groups but “occurrence” (i.e. the percentage of positive preweaned calves is 50%... but only 4 animals were analysed… Can you extrapolate this percentage value to all preweaned calves?).

RESPONSE: Many thanks for your constructive comments. Indeed, only small number of samples were collected from pre/post weaned cattle. As the dairy cattle farmer prefer to fed adult cows that can produce milk than weaned cattle. it is hard to collected enough samples from pre/post weaned cattle in these small-scale farms.

  1. Only seven farms were studied. How was the sampling procedure in each farm? How many samples from each age group were collected in each farm? A careful assessment of your data indicates that no preweaned calves were samples in some farms… It is difficult to compare results between farms if the sampling protocol was different. In addition, the sampling procedure was also different in buffalo farms (the proportion of preweaned calves -0.9 (4/442)- was lower than the proportion of preweaned buffaloes -4.3-, and the proportion of postweaned calves was two times higher -15.2- than that of postweaned buffaloes -7.8). For that reason, you cannot compare prevalences/occurrences.

RESPONSE: Many thanks for your constructive comments and suggestions. All cattle of each farm was the sampling size. Each sample was collected from individual animal from rectum. We have corrected corresponding data in section 3.1.

  1. It was not stated if diarrhoeic animals were sampled. It is also well known that Cryptosporidium infections (especially C. parvum infections) are more frequent in diarrhoeic animals. Epidemiological survey of prevalence of Cryptosporidium spp. in Holstein cows and dairy buffaloes in Yunnan province).

RESPONSE: All cattle were clinically normal without obvious signs of diarrhea. We have corrected corresponding data in Line 135-142.

  1. In addition, although this manuscript provides some new information from a region where data is scarce, results are similar to other previously described in China (and not included in the discussion section). For all these reasons, this manuscript should be significantly improved for being considered for publication in “animals”.

RESPONSE:We thanks for your constructive comments and suggestions. We have added the information in Lines 76-77 and 187-188.

OTHER COMMENTS

  1. Title: in addition to “dairy cattle” add “dairy water buffaloes”

RESPONSE: We have made corresponding revision in the title.

  1. Simple summary should be more accessible for all readers. I think it is still too much technical for all readers.

RESPONSE: We have rewritten the simple summary in Lines 22-28.

  1. Lines 82-82. It is true: little is known about Cryptosporidium infections in cattle from Yunnan. But there is other study (some authors of the present manuscript are also authors of the other investigation) that is not included: Liang et al. (2021). First report of the prevalence and genetic characterization of Giardia duodenalis and Cryptosporidium spp. in Yunling cattle in Yunnan Province, southwestern China, Microbial Pathogenesis,158,105025. https://doi.org/10.1016/j.micpath.2021.105025. Please include and discuss it.

RESPONSE: Many thanks for your reminder. We have added the information in introduction in Lines 76-77 and in discussion in Lines 187-188.

  1. Line 116: delete one of the dots.

RESPONSE: We have made corresponding revision in Line 114.

  1. Sections 2.4 and 2.5: Scientific names should be written in italics.

RESPONSE: We have made corresponding revision in Sections 2.4 and 2.5.

  1. Lines 131-132: Indicate how OR and CI95% were calculated.

RESPONSE: We have added the information in Lines130-131.

  1. Results section: This section should be revised carefully. Firstly, most results (percentages) in the text are repeated in Tables 1 and 2. Do not repeat data on Tables on the text. I recommend keeping percentages on both Tables and indicating patterns or trends in the text. Secondly, be careful with the chi-square results. In some parts it is stated that the differences are significant but they are not. For example in lines 141-144 it is stated that “prevalence in the pre-weaned calves (50.0%) was significantly higher than in post-weaned 142 calves (16.4%), heifers (23.0%), and adult cattle” but this is not completely true. The percentage of infection in preweaned calves was only significantly higher than in adults. The rest were not calculated or not included in the Tables/text. I calculated some and they were not significant. Results in lines 144-146 do not agree with those on the Table 2…  Most differences between age groups are not significant because the number of samples analysed was very small.

In addition, some p values are significant but CI95% of OR values included the number 1… So it should not be significant… Something is wrong. If you are performing a chi-square including all age groups (or considering farms) it will be significant. But if you compare age groups 2 by 2 (i.e. considering the adult group as a reference) most comparisons will not be significant. Be careful since chi-square is not suitable if you have a 2x2 contingency table and some cells have numbers under 5… You should use an exact Fisher test.

For example, if you compare calves (2/4 positives) and adults (43/332), you can see in Table 1 that p= 0.031 and OR was 6.72 (0.92-48.97). Using R:

Your results are obtained like this:

> tabla <- matrix(c(2,2,43,289), nrow = 2)

> chisq.test(tabla, correct = FALSE)

Pearson's Chi-squared test

data:  tabla

X-squared = 4.677, df = 1, p-value = 0.03057

Warning message:

In chisq.test(tabla, correct = FALSE) :

  Chi-squared approximation may be incorrect

> library(vcd)

Loading required package: grid

> oddsratio(tabla, log = FALSE)

 odds ratios for and

[1] 6.72093

> confint(oddsratio(tabla, log = F))

      2.5 %   97.5 %

/ 0.9224279 48.96958

Nevertheless, all this is wrong because you have two cells with less than 5. You must consider that when the number of observations expected for any of the levels is equal to or less than 5, the approximation by the chi square test is not good. Thus, you should use a fisher test (exact test)

> fisher.test(tabla)

Fisher's Exact Test for Count Data

data:  tabla

p-value = 0.08828

alternative hypothesis: true odds ratio is not equal to 1

95 percent confidence interval:

  0.4711809 94.1551091

sample estimates:

odds ratio

  6.656798 

In this case, p is > 0.05, so it is no significant.

RESPONSE: We have recalculated the p value using Fisher's Exact Test, and We have made corresponding revision in section 3.1, table 1 and table 2.

  1. Line 135: it is not 68/700 (overall prevalence) but 65/442. Anyway, these data is on the Table.

RESPONSE: We have made corresponding revision in Line 135.

  1. Lines 137-139: since sampling protocol was very different between farms, comparing overall infection rates has no sense. You can report that the “herd-level occurrence/prevalence” ranged from xx% to xx%.

RESPONSE: We have made corresponding revision in Line 136.

  1. Figures 1 and 2. In my opinion, phylogenetic trees are not needed (what did you use as root of both trees?). The major Cryptosporidium species found are well defined at the 18S gene.

RESPONSE: we have added the root of phylogenetic tree in Fig 2 and we haved deleted the Fig 3.

  1. The discussion section is poor. A predominance of C. andersoni in all age groups in cattle was detected. This is very common in Chinese and in some Brazilian studies on cattle, but very different to the pattern described elsewhere. Please discuss that.

RESPONSE: we have added two reference in line 197.

  1. Lines 202-204. Something is wrong here.

RESPONSE: We have made corresponding revision in Lines 197-200.

  1. IId subtypes are not the most common in cattle worldwide but IIa. Why do you think this C. parvum allelic family is predominant in China? Is this allelic family more frequent in cattle from other countries too?

RESPONSE: Indeed, in most areas, the majority of diarrhoea cases in calves are attributed to the IIa subtypes. Meanwhile, calves in China are commonly infected with IId subtypes, which have caused several outbreaks of cryptosporidiosis in neonatal calves with high mortality. We made corresponding revision in line 211-213.

Round 2

Reviewer 4 Report

Dear authors

The revised version of this manuscript has added interesting information and authors have answered some of my suggestions and comments. In my opinion, and considering that the methods are correct (although a different sampling procedure could increase the impact of the results), the main shortcoming of this manuscript is that data should be better interpreted, and the discussion and conclusions sections must be more informative.

I provide some suggestions for improving the quality and impact of the results:

-English language must be revised, especially in the new parts added, since some little mistakes can be found.

-Simple summary is still difficult to read and understand for people who do not work with Cryptosporidium. I think the first lines of the abstract should be included in the simple summary (Cryptosporidium spp. are important gastrointestinal pathogens of humans and animals, causing diarrheal diseases. Cattle are considered one of the main reservoirs of Cryptosporidium for humans). In addition to first descriptions, you should note the role of asymptomatic cattle and water buffalo on the epidemiology of this protozoan in Yunnan and the Public Health implications of your results. It is a simple summary… include your results and their impact for animal and human health.

-Be careful with the terms prevalence and occurrence, they are not the same. Nevertheless, we can find both throughout the text. If you want to use “occurrence”, please check that no “prevalence” is in the text, and change your objective (line 77). If you want to estimate the prevalence of infection, a significant simple size should be calculated. Generally, some information is needed for calculating the sample size: the estimated true proportion (for example, those obtained in other studies), the population size, the desired precision and the confidence level. Please, check the Epitools web page for sample size calculations (https://epitools.ausvet.com.au/samplesize). Maybe the number of samples included in the study is enough for estimating the prevalence.

-Lines 134-135: It is stated that “Farm Yiliang-2 in Kunming city had significantly higher occurrence rate than other farms (P = 0.00017)”. But, if I am not mistaken, your analysis only allows demonstrating that Farm Yiliang-2 had a significant higher occurrence rate than the Weishan Dali city Farm, which is the reference…

-Line 135: Do not use P= 0.00017. Use P< 0.001.

-Lines 164-165: Were all C. ryanae sequences from cattle buffaloes identical between them? It is not stated.

-Lines 192-195: When discussing the age-related presence of Cryptosporidium species you should state that the number of samples from some age groups in your study is very small and that more robust results should be obtained after analyzing a higher number of samples from pre-weaned, postweaned calves and heifers.

-Line 205: It was stated that this subtype was identified in people (from???), so it has zoonotic potential… But… was the C. parvum subtype IIdA18G1 identified in Chinese people?

-Line 211: change high-toxicity by highly pathogenic.

-The conclusions, especially lines 220 to 223 are results. Please, in this section highlight the importance of your results… What is the importance of detecting a zoonotic subtype of C. parvum in preweaned calves? What is the importance of identifying 4 MLST C. andersoni subtypes? What strategies and measures for controlling Cryptosporidium spp. infection in cattle from Yunnan province would implement considering your results? In addition, further studies, analysing a higher number of samples from young animals will allow having a more complete picture of the prevalence of Cryptosporidium, the species present and their relation with particular age groups. This will allow a better understanding of Cryptosporidium infections in dairy cattle from Yunnan and it will be very interesting to know what age groups are reservoirs of zoonotic or pathogenic species .

Author Response

Response to Reviews for animals-1606913

Thank you very much for positive comments, and constructive suggestions on our manuscript (MS) ID Animals-1606913. We have revised the MS strictly according to the reviewer’ comments and suggestions. We used the “tracked changes” mode in the WORD to show the revised/changed text in the revised MS. Two MS files are uploaded: the “clean version” as “manuscript”, and the one showing “tracked changes”. In the following, we detail our point-by-point responses to the reviewer’s comments and suggestions. We made all our responses in blue color for clarity. The following line numbers correspond to those in the marked copy of the revised manuscript.

Responses to reviewer #4

Comments and Suggestions for Authors

Dear authors

The revised version of this manuscript has added interesting information and authors have answered some of my suggestions and comments. In my opinion, and considering that the methods are correct (although a different sampling procedure could increase the impact of the results), the main shortcoming of this manuscript is that data should be better interpreted, and the discussion and conclusions sections must be more informative.

I provide some suggestions for improving the quality and impact of the results:

RESPONSE: We thanks reviewer #4 very much for the constructive comments and suggestions.

  1. English language must be revised, especially in the new parts added, since some little mistakes can be found.

RESPONSE: Thank you for your careful tip, we have made corresponding revision about the English language in lines 79, 91,96,140,146,163.

  1. Simple summary is still difficult to read and understand for people who do not work with Cryptosporidium. I think the first lines of the abstract should be included in the simple summary (Cryptosporidium spp. are important gastrointestinal pathogens of humans and animals, causing diarrheal diseases. Cattle are considered one of the main reservoirs of Cryptosporidium for humans). In addition to first descriptions, you should note the role of asymptomatic cattle and water buffalo on the epidemiology of this protozoan in Yunnan and the Public Health implications of your results. It is a simple summary… include your results and their impact for animal and human health.

RESPONSE: Many thanks for your constructive comments and suggestions. We have rewritten the simple summary in lines 23-32.

  1. Be careful with the terms prevalence and occurrence, they are not the same. Nevertheless, we can find both throughout the text. If you want to use “occurrence”, please check that no “prevalence” is in the text, and change your objective (line 77). If you want to estimate the prevalence of infection, a significant simple size should be calculated. Generally, some information is needed for calculating the sample size: the estimated true proportion (for example, those obtained in other studies), the population size, the desired precision and the confidence level. Please, check the Epitools web page for sample size calculations (https://epitools.ausvet.com.au/samplesize). Maybe the number of samples included in the study is enough for estimating the prevalence.

RESPONSE: Many thanks for your constructive comments and suggestions. We have used the “occurrence” to replace the “prevalence” which throughout the text.

  1. Lines 134-135: It is stated that “Farm Yiliang-2 in Kunming city had significantly higher occurrence rate than other farms (P = 0.000017)”. But, if I am not mistaken, your analysis only allows demonstrating that Farm Yiliang-2 had a significant higher occurrence rate than the Weishan Dali city Farm, which is the reference…

RESPONSE: Many thanks for your constructive comments and suggestions. We have made corresponding revision in lines 139-140. As the highest infection rate was 38% in farm Yiliang-2, which was significantly higher than that in farm Eryuan (13.3%, p =0.0027) and other farms, so we use P< 0.01 to replace P = 0.000017.

  1. Line 135: Do not use P= 0.00017. Use P< 0.001.

RESPONSE: We have made corresponding revision in line 140. we use P< 0.01 to replace P = 0.00017.

  1. Lines 164-165: Were all C. ryanae sequences from cattle buffaloes identical between them? It is not stated.

RESPONSE: Many thanks for your constructive comments and suggestions. We have added the information in introduction in lines 169-172.

  1. Lines 192-195: When discussing the age-related presence of Cryptosporidium species you should state that the number of samples from some age groups in your study is very small and that more robust results should be obtained after analyzing a higher number of samples from pre-weaned, postweaned calves and heifers.

RESPONSE: Many thanks for your constructive comments and suggestions.We have rewritten the section and added one reference in lines 199-205.

  1. Line 205: It was stated that this subtype was identified in people (from???), so it has zoonotic potential… But… was the C. parvum subtype IIdA18G1 identified in Chinese people?

RESPONSE: Many thanks for your constructive comments and suggestions.We have rewritten the section and added three references in lines 217, 223-224.

  1. Line 211: change high-toxicity by highly pathogenic.

RESPONSE: we have made corresponding revision in line 221.

  1. The conclusions, especially lines 220 to 223 are results. Please, in this section highlight the importance of your results… What is the importance of detecting a zoonotic subtype of C. parvum in preweaned calves? What is the importance of identifying 4 MLST C. andersoni subtypes? What strategies and measures for controlling Cryptosporidium sppinfection in cattle from Yunnan province would implement considering your results? In addition, further studies, analysing a higher number of samples from young animals will allow having a more complete picture of the prevalence of Cryptosporidium, the species present and their relation with particular age groups. This will allow a better understanding of Cryptosporidium infections in dairy cattle from Yunnan and it will be very interesting to know what age groups are reservoirs of zoonotic or pathogenic species.

RESPONSE: Many thanks for your constructive comments and suggestions.We have rewritten the section of conclusions in lines 228-239.
